# The Variation of Soil Phosphorus Fractions and Microbial Community Composition under Consecutive Cucumber Cropping in a Greenhouse

Ting Bian [1,2,3,4], Shiwei Zheng [1,2,3,4], Xiao Li [1,2,3,4], Shuang Wang [1,2,3,4], Xiaolan Zhang [1,2,3,4], Zhen Wang [1,2,3,4], Xiaoxia Li [1,2,3,4], Hongdan Fu [1,2,3,4,*] and Zhouping Sun [1,2,3,4,*]

1   College of Horticulture, Shenyang Agricultural University, Shenyang 110866, China; bbdbb17637035@163.com (T.B.); zhengzhengzsw@126.com (S.Z.); xiao18240159945@163.com (X.L.); ws5211006@163.com (S.W.); zxl82fighting@163.com (X.Z.); whw19941222@163.com (Z.W.); leexx0911@163.com (X.L.)
2   Key Laboratory of Protected Horticulture of the Education Ministry and Liaoning Province, College of Horticulture, Shenyang Agricultural University, Shenyang 110866, China
3   National & Local Joint Engineering Research Center of Northern Horticultural Facilities Design & Application Technology (Liaoning), Shenyang 110866, China
4   Collaborative Innovation Center of Protected Vegetable Surround Bohai Gulf Region, Shenyang 110866, China
*   Correspondence: fuhongdan@syau.edu.cn (H.F.); sunzp@syau.edu.cn (Z.S.)

**Abstract:** The distribution of phosphorus (P) fractions in soil plays a decisive role in soil P bioavailability; however, the characteristics of soil P fractions under consecutive cropping in a solar greenhouse remain unclear. To evaluate the effects of the long-term successive vegetable cropping on soil P fractions and the microbial community composition in greenhouse soil, a continuous long-term cropping experiment was conducted using cucumber (*Cucumis sativus* L.) in a solar greenhouse starting from 2006 to 2018. Soil P fractions and the microbial community composition were determined using the Hedley continuous extraction method and the phospholipid fatty acid (PLFA) method, respectively, in the 1st, 9th, 13th, and 21st rounds of cultivation. The soil total phosphorus (TP) content increased from 0.90 g·kg$^{-1}$ in the 1st round to 3.07 g·kg$^{-1}$ in the 21st round of cucumber cultivation. With an increase in continuous cropping rounds, soil available phosphorus (AP) increased and the phosphorus activation coefficient (PAC) decreased, with no significant difference between the 13th and 21st rounds. After 21 rounds of continuous cropping, the soil organic matter (SOM) content was 16.34 g·kg$^{-1}$, 1.42 times that of the 1st round. The abundance of soil bacteria, actinomycetes, Gram-negative bacteria (G$^-$), Gram-positive bacteria (G$^+$), and total PLFAs initially increased with continuous cropping rounds, but then decreased significantly, and the ratios of fungi:bacteria (F/B) and G$^+$/G$^-$ bacteria also increased significantly with continuous cropping rounds. The contents of soil labile P, moderately labile P, and non-labile P increased significantly over 21 continuous cropping rounds. Together, these results demonstrate that long-term continuous cropping can directly lead to the accumulation of P fractions, but it can also affect the abundance of actinomycetes through SOM enrichment, which indirectly leads to the accumulation of non-labile P. This study provides a theoretical basis for future soil P fertilizer management and vegetable production sustainability.

**Keywords:** greenhouse; cucumber successive cropping; soil P fractions; soil microbial community composition



## 1. Introduction

Phosphorus (P) is one of the indispensable nutrients in plant growth and development [1]. An over-abundance or deficiency in the P supply directly affects the growth and development of a crop [2]. Under low P stress, the underdeveloped root system of plants leads to slow plant growth and decreased yield. Excessive P may lead to crop metabolism

disorders, quality decline, and an increase in the risk of P loss due to leaching [3]. P fertilizer is widely used to promote crop growth and increase crop yield. Phosphate rock, a non-renewable resource, is the main source of phosphate fertilizer production [4]. It is estimated that the existing global phosphate reserves will last 50–100 years, while in China, the phosphate reserves will be exhausted by 2050 [5]. Therefore, studies that explore the changing of soil P have important guiding significance for the rational application of P fertilizer in the future.

In the past twenty years, greenhouse vegetable production (GVP) has rapidly developed in China, where it plays an essential role in meeting the need for fresh vegetables, especially during the winter months [6]. In China, GVP has long been carried out under conditions of semi-closed high temperature, humidity, water, and fertilizer, which has led to the observation that the P enrichment of greenhouse soil is generally higher than that of field soil [7,8]. Inorganic P ($P_i$) and organic P ($P_o$) are the main forms of soil P, and soil $P_o$ can be absorbed and utilized by plants only after being mineralized to $P_i$. The Hedley P fractional extraction method revised by Tissen has been widely adopted by scholars. Using this method and based on plant availability soil P to plants, it can be divided into labile P (resin-P, $NaHCO_3$-$P_i$, and $NaHCO_3$-$P_o$), moderately labile P ($NaOH$-$P_i$, $NaOH$-$P_o$, and Dil. $HCl$-$P_i$), and non-labile P (conc. $HCl$-$P_i$, conc. $HCl$-$P_o$, and residual-P) [9]. Labile P is the most easily absorbed and utilized by plants [10]. Moderately labile P refers to the P that is absorbed on the surface of soil iron (Fe), aluminum (Al) compounds, and clay particles by chemical adsorption [11]. Non-labile P has a slow turnover in soil and is difficult to be utilized by plants. After 28 years of wheat cultivation, P accumulation was mainly via non-labile P [12]. Previous studies characterizing P fractions focused on continuous cropping soils in the field; however, vegetable crops grown in a greenhouse grow faster and have a shorter growth cycle than that of field crops, requiring more P fertilizer. Therefore, it is important to explore the changes in P fractions in greenhouse soil under long-term continuous cropping to reveal the nutrient status and transformation of P in soil.

Soil microbes play an essential role in the material circulation of the ecosystem [13]. Bacteria and fungi are the most important two groups of soil microorganisms, and changes in them represent soil biological activity [14,15]. Long-term continuous cropping typically leads to a decrease in the abundance of soil bacteria and an increase in fungi abundance. Total phosphorus (TP) and available phosphorus (AP) are positively correlated with bacteria, whereas both are negatively correlated with fungi [16]. Gram-positive bacteria ($G^+$) were positively correlated with soil residual-P in maize after 60 years of cultivation [17]. However, there are few studies on the correlation between phospholipid fatty acid (PLFA) content and P fractions in long-term GVP planting systems.

Soil microbial community composition and its impact on soil P cycling under long-term consecutive cucumber (*Cucumis sativus* L.) cropping in a greenhouse environment are still largely unknown. The objectives of this study are to (1) investigate the characteristics of soil P fractions in cucumber continuously cultivated for 21 rounds; (2) clarify the changes in soil microbial community composition characteristics; and (3) explore the relationships between the characteristics of the soil P fractions, soil microbial community structure, and soil chemical properties.

## 2. Materials and Methods

### 2.1. Site Description and Experimental Design

The experiment was conducted in a solar greenhouse located at Shenyang Agriculture University, Liaoning Province, China (41.31° N, 123.240° E) and the cucumber variety tested was "Jinyou 30". The original soil used for continuous cropping was selected from nearby fallow land, previously under maize cultivation, classified as Hapli-Udic Cambisols. The basic chemical properties and P fraction content of the original soil are shown in Table 1.

**Table 1.** Basic properties and P fraction content of the tested materials.

| Materials | pH | EC | TC | TN | TP | TK | AN | AP | AK |
|---|---|---|---|---|---|---|---|---|---|
| | | $ms \cdot cm^{-1}$ | $g \cdot kg^{-1}$ | $g \cdot kg^{-1}$ | $g \cdot kg^{-1}$ | $g \cdot kg^{-1}$ | $mg \cdot kg^{-1}$ | $mg \cdot kg^{-1}$ | $mg \cdot kg^{-1}$ |
| | 7.5 | 0.2 | 12.0 | 1.3 | 0.9 | 10.6 | 102.0 | 41.3 | 222.8 |
| Original soil | Resin-P | $NaHCO_3$-$P_i$ | NaOH-$P_i$ | Dil. HCl-$P_i$ | Conc. HCl-$P_i$ | $NaHCO_3$-$P_o$ | NaOH-$P_o$ | Conc. HCl-$P_o$ | Residual-P |
| | $mg \cdot kg^{-1}$ | $mg \cdot kg^{-1}$ | $mg \cdot kg^{-1}$ | $mg \cdot kg^{-1}$ | $mg \cdot kg^{-1}$ | $mg \cdot kg^{-1}$ | $mg \cdot kg^{-1}$ | $mg \cdot kg^{-1}$ | $mg \cdot kg^{-1}$ |
| | 92.0 | 16.3 | 43.9 | 233.8 | 42.0 | 14.8 | 18.9 | 3.1 | 14.1 |

EC, electrical conductivity; TC, total carbon; TN, total nitrogen; TP, total phosphorus; TK, total potassium; AN, available nitrogen; AP, available phosphorus; and AK, available potassium.

The experiment was initiated in 2006. At the beginning of September 2006, 2007, 2009, and 2010 and at the end of February 2012, 2013, 2014, 2015, 2016, 2017, and 2018, field soil (440 kg dry weight) from the same site near the experimental greenhouse was placed in cultivation tanks (315 cm long × 60 cm wide × 30 cm high) located in a greenhouse (Figure 1). Owing to the relocation of the greenhouse base, the addition of soil to the greenhouse cultivation tanks in September 2011 was postponed to February 2012. To separate the added soil from the native soil under the tanks, a plastic film with 2 rows of drainage holes (2 cm in diameter) was laid on the bottom of each plot (that is, the plot depth = 0.3 m). The experiment consisted of a random block design; each cultivation tank was a plot, with three repetitions, and continuous monoculture in the spring (March–June) and autumn (September–December) each year. After comparing the growth and yield of the cucumber plants at each time point over the course of the experiment, soil from the 1st, 9th, 13th, and 21st rounds of cultivation was selected for testing and analysis in June 2018.

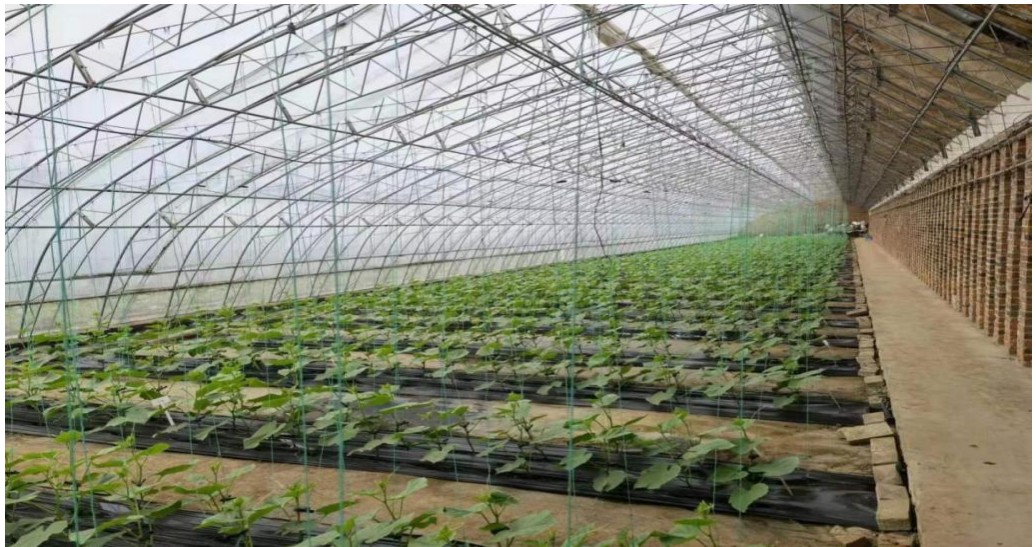

**Figure 1.** Test images.

Before cucumber planting, 4.5 kg of commercial chicken manure was applied to each cultivation tank. The basic chemical properties of the chicken manure were total carbon (TC), 295 $g \cdot kg^{-1}$; total nitrogen (TN), 51.4 $g \cdot kg^{-1}$; total phosphorus (TP), 21.6 $g \cdot kg^{-1}$; and total potassium (TK), 35.4 $g \cdot kg^{-1}$. Cucumber seedlings with three leaves were planted in the cultivation tank in March 2018. Each tank was cultivated in 2 rows (8 plants per row) with a plant spacing of 0.5 m × 0.4 m. A drip irrigation belt was used for irrigation. Sown seedlings were watered 7 days after planting, and fertilization was begun at a fixed value after 20 days, using a total of 8 applications. The specific fertilizer content of nitrogen (N), phosphorus pentoxide ($P_2O_5$), and potassium oxide ($K_2O$) (added eight times) is shown in Table 2.

**Table 2.** Fertilizers and manure applied to the greenhouse cucumber cropping systems.

| Elemental Sources | Elemental Application Amount (kg·ha$^{-1}$ year$^{-1}$) | | |
|---|---|---|---|
| | N | P$_2$O$_5$ | K$_2$O |
| Chemical fertilizer | 450 | 142 | 593 |
| Chicken manure | 361 | 152 | 249 |
| Total fertilizer | 812 | 294 | 842 |

### 2.2. Soil Sampling

At the beginning of June 2018, 5 cucumber soil samples at a depth of 0–20 cm were randomly selected from each cultivation tank using a soil collector, and a final soil sample was formed after mixing the 5 samples from each tank. Each mixed soil sample was then screened with a 2 mm sieve to remove any debris and subsequently mixed evenly. Then, each soil sample was sub-divided and treated as follows: (1) air-dried for soil chemical analysis and P fractionation, or (2) stored fresh at 4 °C for enzyme activity and PLFA determination.

### 2.3. Chemical Analyses in Soil and Plants

Each soil sample was also analyzed for pH, electrical conductivity (EC), soil organic matter (SOM), TP, TK, AP, available potassium (AK), and phosphorus activation coefficient (PAC). Soil pH and EC were measured in a soil:water suspension (1:5 m/v) after 30 min of shaking at 28 °C, and using a pH meter (PB-10, Sartorius, Goettingen, Germany) and conductivity meter (DDS-307A, INESA Scientific Instrument, Shanghai, China), respectively [18]. The SOM was determined by potassium dichromate titration [18]. Soil TP and TK were measured using 0.2 g (100 mesh) soil samples, to which 5 mL HNO$_3$, 2 mL HF, and 2 mL HCl were successively added in Teflon vessels. The vessels were placed in a microwave digestion facility (MARS6, CEM Corporation, Matthews, CA, USA) and digested using a two-step digestion procedure. TP was determined by using by molybdenum blue colorimetry and TK by using atomic absorption spectrometry (iCE3000, Thermo Fisher Scientific, Waltham, MA, USA). Soil AN was determined using the alkali-hydrolyzed diffusion method. AP was determined by extraction with 0.5 M NaHCO$_3$ and measured using the molybdenum blue spectrophotometry method [19]. Soil AK was extracted using 1 M NH$_4$OAc and determined using atomic absorption spectrometry (iCE3000, Thermo Fisher Scientific). PAC was quantified from the ratio of AP to TP in the soil sample. Total P in the plants was extracted using the H$_2$SO$_4$-H$_2$O$_2$ method, and the concentration of P in the extracts was determined using the molybdenum blue colorimetric method [18].

### 2.4. Soil Phosphorus Fractionation and Phosphatase Activity

A modified Hedley continuous extraction method was used to determine the soil P fractions [20,21]. Briefly, 0.5 g of a 100-mesh soil-sieved sample was transferred to a 50 mL centrifuge tube and 30 mL of deionized water was added to extract resin-P. The other fractions were successively extracted in other using 0.5 M NaHCO$_3$ pH 8.5, (NaHCO$_3$-P), 0.1 M NaOH (NaOH-P), and 1 M HCl (HCl-P). After each addition, the soil-liquid mixture was shaken for 16 h at 25 °C at 250 rpm and centrifuged at 10,000 rpm for 10 min. The supernatant was collected and filtered through a 0.45 μm membrane filter. The residual-P was digested with concentrated H$_2$SO$_4$-H$_2$O$_2$. The molybdate colorimetric method was used to determine the total inorganic phosphorus (TP$_i$) and TP. Soil total organic P (TP$_o$) in all extracts was calculated as the difference between TP and TP$_i$.

The Tabatabai method was used to determine soil phosphomonoesterase activity and phosphodiesterase activity [22]. As the pH value of the soil was high, the alkaline phosphomonesterase activity of the soil was measured. The activity of alkaline phosphatase was extracted using a buffer solution at pH 11.0, and its activity spectrophotometrically determined at 410 nm using p-nitrobenzene sodium phosphate as the substrate. Soil

phosphodiesterase activity was determined by a similar method, except that a buffer at pH 8.0 was used with a solution of sodium nitrobenzene phosphate as the substrate.

### 2.5. Soil of Phospholipid-Derived Fatty Acids (PLFAs)

The structural characteristics of the soil microbial communities were determined using a PLFA method [23]. Extraction was carried out using the KOH-methanol solution methyl ester method; 19:0 methyl ester was added to the esterified samples as an internal standard. An Agilent GC 7890B gas chromatograph was used for fatty acid comparison and identification using the Sherlock Microbial Identification System (MIS) developed by the MIDI Company. The $^{13}$C value of each fatty acid was determined using a gas chromatography combustion isotope mass spectrometer (GC-IRMS, MAT 253, Thermo Fisher). The characterization of the microbial PLFAs is shown in Table 3.

**Table 3.** Phospholipid fatty acid (PLFA) characterization of microorganisms.

| Microbial Type | Phospholipid Fatty Acid Labelled |
|---|---|
| Gram-positive bacteria (G$^+$) | 17:0 iso, 17:0 anteiso, 16:0 iso, 15:0 iso, 15:0 anteiso, 14:0 iso |
| Gram-negative bacteria (G$^-$) | 16:1 w7c, 16:1 w9c, 17:0 cyclo w7c, 17:1 w8c, 18:1 w7c, 19:0 cyclo w7c |
| Actinomycetes | 18:0, 17:0 10-methyl, 16:0 10-methyl, 18:0 10-methyl |
| Fungi | 18:2 w6c, 18:1 w9c, 16:1 w5c |

### 2.6. Statistics Analysis

All statistical analyses were performed using SPSS 22.0 (SPSS Inc., Chicago, IL, USA). One-way ANOVA and Duncan's test method were used for the analysis of variance and multiple comparison ($p < 0.05$). Pearson correlation analysis was used to determine the correlation between soil properties or microbial community structure and P fractions, and conducted in Origin Pro 2022. Principal Component Analysis (PCA) with the Monte Carlo permutation test was performed to determine the relationship among soil properties or microbial community composition and soil P fractions, and implemented in Origin Pro 2022. AMOS software (IBM SPSS AMOS 24.0.0) was used to construct a structural equation model (SEM) to illustrate the relationship between soil chemical properties, P fraction, and microbial community composition under long-term continuous cropping. Using the goodness of fit index CFI (>0.95) and root-mean-square error of approximation (RMSEA), a value of 0.05 indicated a good fit; if RMSEA = 0, the model was completely fitted. The chi-squared test of goodness of fit ($\chi^2$) was used together with degree of freedom (DF); the closer $\chi^2$/DF is to 1, the better the model is fitted.

## 3. Results

### 3.1. Uptake and Distribution of P in Plants

In comparison to the 1st round of analysis, P uptake by cucumber plants increased significantly over 21 continuous cropping rounds. There was no significant difference in the P-uptake content of cucumber roots, leaves, and fruits among each continuous cropping rounds (Table 4). The distribution of P accumulation in all organs was leaf > fruit > stem. The P distribution was 36.53–48.96% in the leaf, 31.85–49.09% in the fruit, 12.96–20.09% in the stem, and 1.20–1.56% in the root, among which, P accumulation in the fruits in the 13th round was higher than that in the leaves; in the other rounds, P accumulation in the leaves was higher than that in the fruits.

**Table 4.** Effects of continuous cropping on P uptake and distribution of cucumbers in the solar greenhouse.

| Treatments | P Accumulation (g·Plant⁻¹) | | | | | P Distribution Proportion (%) | | | |
|---|---|---|---|---|---|---|---|---|---|
| | Root | Stem | Leaf | Fruit | Total | Root | Stem | Leaf | Fruit |
| 1 | 0.0068 ± 0.0004 a | 0.05 ± 0.01 b | 0.21 ± 0.02 a | 0.19 ± 0.02 a | 0.45 ± 0.02 b | 1.50 ± 0.16 a | 10.49 ± 1.08 c | 47.15 ± 3.04 a | 40.86 ± 3.70 a |
| 9 | 0.0080 ± 0.0010 a | 0.11 ± 0.01 a | 0.22 ± 0.01 a | 0.19 ± 0.01 a | 0.52 ± 0.03 ab | 1.56 ± 0.25 a | 20.09 ± 1.52 a | 42.59 ± 2.08 a | 35.76 ± 0.74 a |
| 13 | 0.0084 ± 0.0010 a | 0.08 ± 0.01 ab | 0.21 ± 0.02 a | 0.29 ± 0.04 a | 0.59 ± 0.01 a | 1.42 ± 0.15 a | 12.96 ± 2.30 bc | 36.53 ± 3.51 a | 49.09 ± 5.64 a |
| 21 | 0.0068 ± 0.0008 a | 0.11 ± 0.01 a | 0.29 ± 0.06 a | 0.18 ± 0.05 a | 0.58 ± 0.04 a | 1.20 ± 0.21 a | 17.99 ± 1.22 ab | 48.96 ± 8.80 a | 31.85 ± 10.02 a |

Different letters indicate significant difference among the treatments on $p < 0.05$.

### 3.2. Soil Chemical Properties

In comparison to the 1st round, the EC, TK, AN, and AK in the soil increased significantly by 39.70%, 45.29%, 218.92%, and 218.92%, respectively, after 21 continuous cropping rounds (Table S1). Except for AK, there was no significant difference between the 9th, 13th, and 21st rounds. In the soil, there was no significant difference in pH among the different rounds.

TP and AP in the soil showed a significant increasing trend from the 1st to the 21st continuous cropping round (Figure 2). The accumulation of soil TP and AP increased from 0.9 g·kg⁻¹ to 3.07g·kg⁻¹ and 82.68 mg·kg⁻¹ to 200.19 mg·kg⁻¹, respectively, over 21 continuous cropping rounds. However, in the 13th and 21st rounds, there was no significant difference in the soil AP amount. In comparison to the 1st round, the PAC of the 13th and 21st rounds decreased significantly by 28.16% and 31.36%, respectively, in the soil.

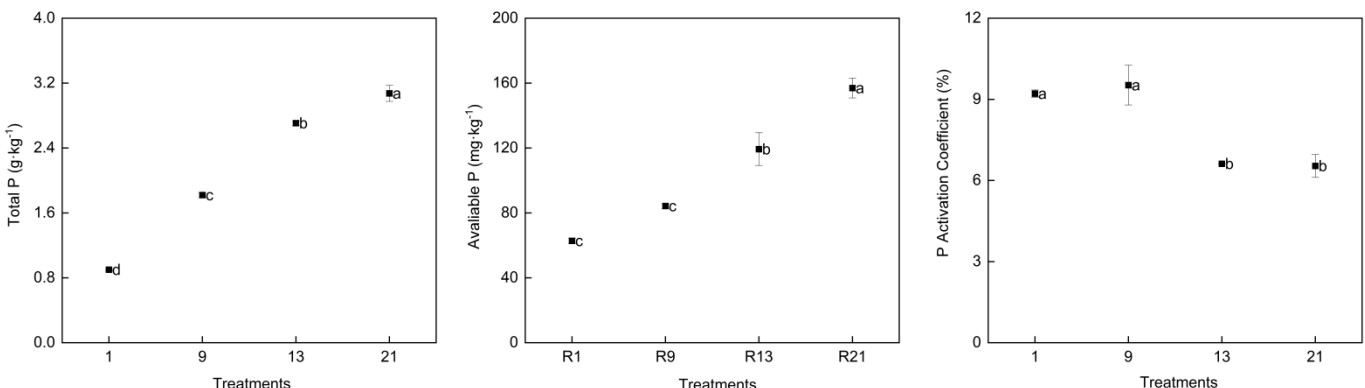

**Figure 2.** Changes of total P, available P, and P activation coefficient in the greenhouse soil after long-term continuous cropping. The different letters indicate the significant differences among the treatments on $p < 0.05$. P activity coefficient: phosphorus activity coefficient.

### 3.3. Soil Phosphorus Fractions

After long-term continuous cropping, the soil P fraction of the greenhouse cucumber was mainly $P_i$, accounting for 84.06–89.81% of TP, and $P_o$ accounted for 10.19–15.94% of the TP (Table S2). The ratio of $P_i$/TP in late continuous cropping (13th and 21st rounds) was significantly lower than that in early continuous cropping (1st and 9th rounds), and the trend for $P_o$/TP was the opposite.

With an increase in the continuous cropping rounds, the accumulation of labile P and moderately labile P in the soil showed a significant increasing trend (Figure 3). After 21 rounds of planting, soil labile P and moderately labile P increased by 260.44% and 355.93%, respectively, compared to that measured in the 1st round. There was also no significant difference between the 9th and 13th rounds. The content of non-labile P increased significantly from 158.90 mg·kg⁻¹ to 256.94 mg·kg⁻¹ after the 1st round, but there was no significant difference among the content in the 9th, 13th, and 21st rounds. There was also no significant difference in the ratio of soil labile P to TP in each continuous cropping round (Table S3). After long-term continuous cropping, the proportion of moderately labile P in the soil increased significantly. In comparison to the 1st round, it increased by

16.85%, 20.13%, and 20.37% in the 9th, 13th, and 21st rounds, respectively, but there was no significant difference among the 3 continuous cropping rounds.

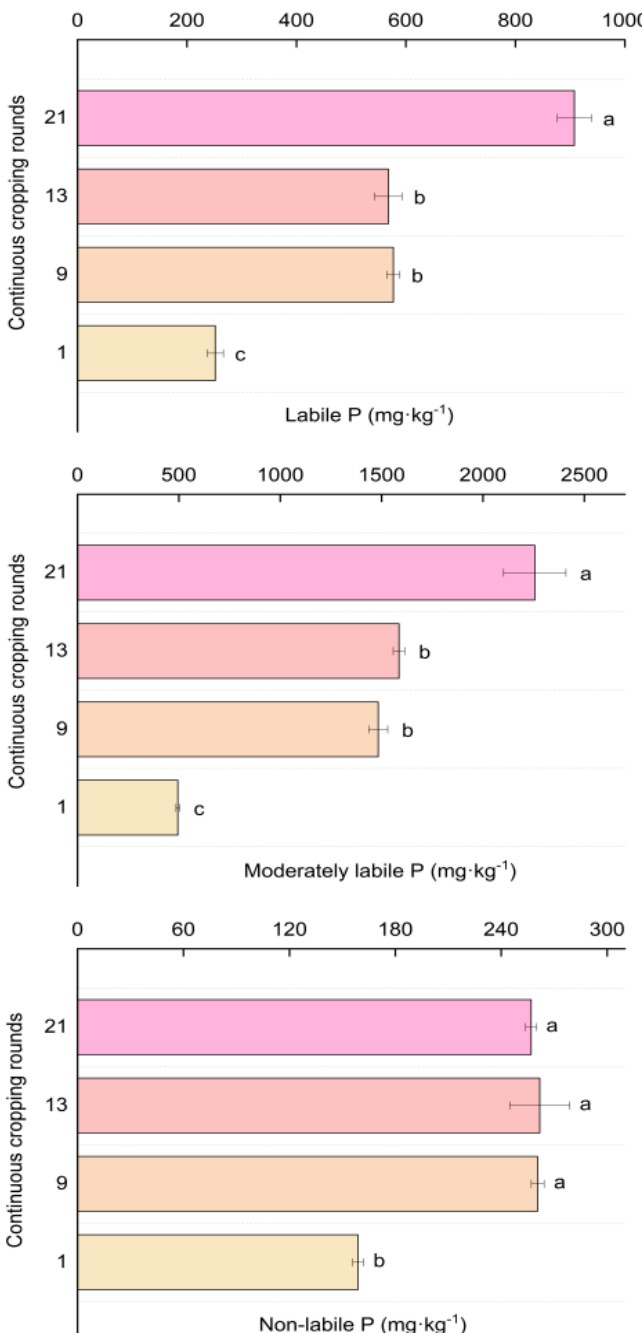

**Figure 3.** Changes of P fraction contents in the greenhouse soil after long-term continuous cropping. The different letters indicate the significant differences among the treatments on $p < 0.05$.

In this study, the continuous cropping rounds had no significant effect on the proportion of soil resin-P and $NaHCO_3$-$P_o$ in total P, but the proportion of soil $NaHCO_3$-$P_i$ in total P decreased significantly from the 1st to the 21st cropping round (Figure 4).

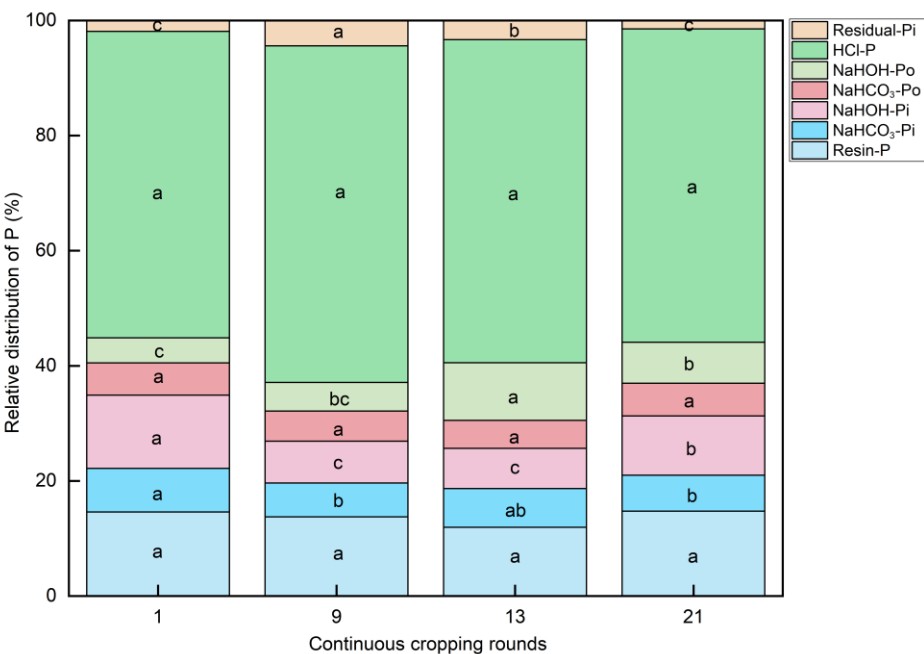

**Figure 4.** The relative distribution of P fraction in 0–20 cm soil under long-term continuous cropping. The different letters indicate the significant differences among the treatments on *p* < 0.05.

In comparison to the proportion of NaOH-$P_i$/TP in the 1st round, the proportion in the 9th, 13th, and 21st rounds decreased significantly. The values of the ratio in the 9th and 13th round were 7.93% and 7.49%, respectively, which were the lowest (*p* < 0.05). The proportion of NaOH-$P_o$/TP increased significantly from the 1st to 13th cropping round, and reached its highest value (10.01%) in the 13th round, followed by a decreasing trend. The proportion of dil. HCl-$P_i$/TP in the 1st round was the lowest (*p* < 0.05), but there was no significant difference between the 9th, 13th, and 21st rounds (Table S4). The proportion of conc. HCl-$P_i$/TP decreased significantly from 14.79% after the 1st round to 4.03% after the 21st round (Table S4). The proportion of NaHCO$_3$-$P_o$ and conc. HCl-$P_o$ in TP$_o$ initially increased and then decreased, with an increase in the number of planting rounds. NaOH-$P_o$ accounted for 62.73% of the TP$_o$ in the 13th round (Table S5).

### 3.4. Soil Phosphatase Activity and Soil Organic Matter

Both phosphomonoesterase and phosphodiesterase activity initially increased and then decreased (Figure 5). There was no significant difference in the phosphomonoesterase content between the 13th and 21st rounds, but the phosphodiesterase content of the 13th rounds was 7.58 mg·g$^{-1}$·soil·h$^{-1}$, which was significantly higher than that measured in the 21st round. SOM content increased significantly, with the 9th, 13th, and 21st rounds being 1.32-, 1.27-, and 1.42 times that of the 1st round, respectively.

### 3.5. Soil Microbial Community Composition

The dynamic changes in the soil microbial community composition in the greenhouse cucumber continuous cropping system are shown in Table 5. The content of soil bacteria, actinomycetes, G$^+$, G$^-$, and total PLFAs show an increasing trend from the 1st to the 9th rounds, reaching a peak in the 9th round, and then show a downward trend. There is no significant difference between the 9th and 13th rounds, and the 21st round is significantly lower than the 9th round. After 21 continuous cropping rounds of planting, the soil F/B and G$^+$/G$^-$ ratios reach their highest levels (*p* < 0.05), which are 0.22 and 0.88, respectively.

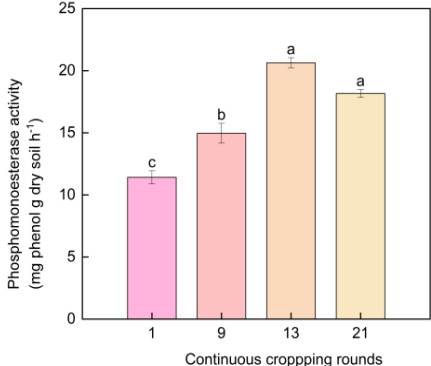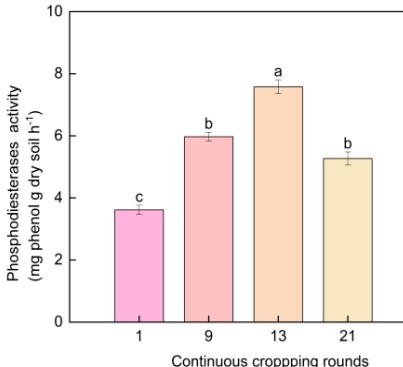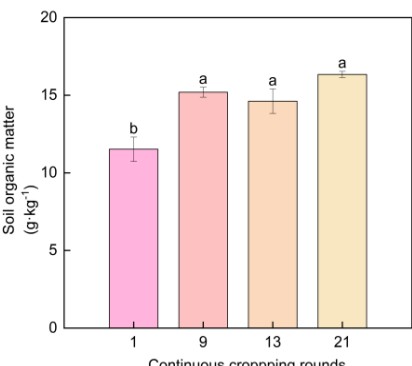

**Figure 5.** Effects of different continuous cropping rounds on soil phosphomonoesterase, phosphodiesterase activity, and soil organic matter for the greenhouse soil. The different letters indicate the significant differences among the treatments on $p < 0.05$.

**Table 5.** Changes of soil percentage of phospholipid fatty acids after long-term continuous cropping.

| Treatments | Bacteria ug·g$^{-1}$ | Fungi ug·g$^{-1}$ | Actinomycetes ug·g$^{-1}$ | G$^+$ ug·g$^{-1}$ | G$^-$ ug·g$^{-1}$ | F/B | G$^+$/G$^-$ | Total PLFA ug·g$^{-1}$ |
|---|---|---|---|---|---|---|---|---|
| R1 | 33.00 ± 3.12 b | 5.12 ± 0.74 b | 5.91 ± 0.42 c | 13.46 ± 1.55 c | 18.64 ± 1.58 b | 0.15 ± 0.01 b | 0.72 ± 0.02 b | 48.62 ± 5.08 b |
| R9 | 60.11 ± 3.17 a | 10.48 ± 0.54 a | 11.28 ± 0.58 a | 25.51 ± 1.25 a | 32.87 ± 2.06 a | 0.17 ± 0.01 b | 0.78 ± 0.04 b | 88.06 ± 4.95 a |
| R13 | 57.79 ± 4.36 a | 8.89 ± 0.43 a | 11.43 ± 0.53 a | 24.95 ± 1.59 a | 31.35 ± 2.64 a | 0.16 ± 0.01 b | 0.80 ± 0.02 b | 86.33 ± 8.87 a |
| R21 | 41.18 ± 2.59 b | 8.33 ± 0.92 a | 9.14 ± 0.57 b | 18.78 ± 1.38 b | 21.38 ± 1.29 b | 0.22 ± 0.02 a | 0.88 ± 0.01 a | 63.13 ± 4.01 b |

The different letters indicate the significant differences among the treatments on $p < 0.05$. The values are mean ± standard errors. G$^-$, Gram-negative bacterium; G$^+$, Gram-positive bacterium.

### 3.6. The Relationships between PLFA, Soil Parameters, and P Fractions

PCA analysis was used to study the effects of different planting rounds on soil microbial community composition and P fractions. PCA yielded two principal components, which contributed most to the change in soil properties and could therefore be used for further interpretation of the results. According to Figure 6a, PC1 and PC2 contributed 94.1% and 4.1%, respectively, to the change in PLFAs in the soil. The samples were divided into three different groups, each representing a different continuous cropping time. The first group consisted of the 1st round, i.e., soil with a low continuous cropping time. The second group consisted of the 9th and 13th rounds. The third group consisted of the 21st round, i.e., the longest continuous cropping time. The content of soil P was analyzed by PCA, showing that PC1 and PC2 contributed 89.6% and 9.1% to the change in soil P fractions, respectively (Figure 6b). The results show that the soil P fraction of short-term continuous cropping and long-term continuous cropping can be clearly distinguished along the χ axis.

In general, soil EC, AN, and SOM were significantly positively correlated with labile P and moderately labile P. EC was significantly positively correlated with CHCl-P$_O$, and AN was significantly positively correlated with residue-P (Figure 7). Actinomycetes were positively correlated with moderately labile P and non-labile P. Residual-P$_i$ was significantly affected by PLFA (Figure S1).

An SEM model was used to elucidate the relationships between soil P fraction, chemical properties, and microbial community composition under long-term continuous cropping. The model fitting results were $\chi^2 = 6.423$, DF = 7, CFI = 1.000, AIC = 46.423, and RMSEA = 0.000, indicating that the model fit well. Long-term single continuous cropping itself led to the accumulation of soil labile P, moderately labile P, and non-labile P. Continuous cropping also significantly affected the soil SOM content, and SOM had a significant positive effect on the abundance of actinomycetes, which indirectly promoted the accumulation of non-labile P in the soil (Figure 8).

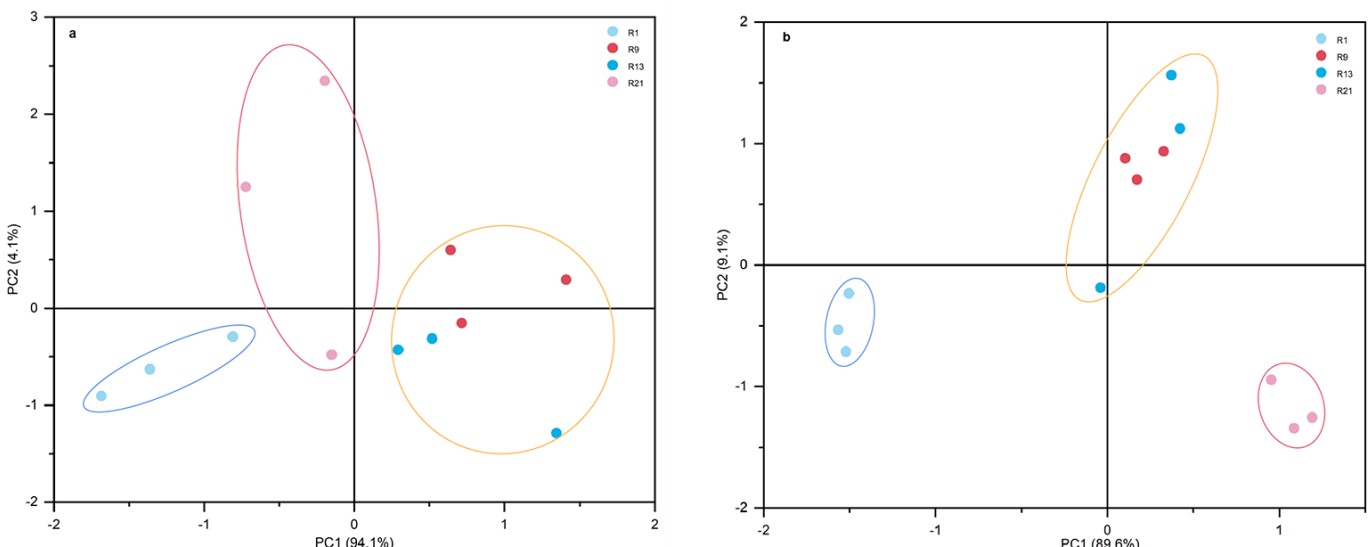

**Figure 6.** The principal component analysis (PCA) of the microbial community composition in soils (**a**) and P fraction (**b**) under long-term continuous cropping.

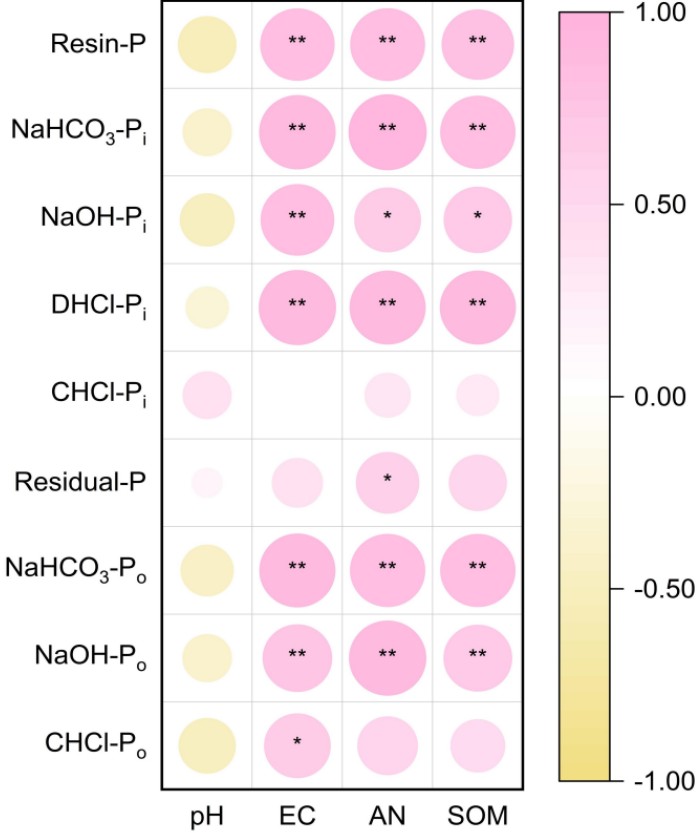

**Figure 7.** The Pearson correlation analysis for soil chemical properties and P fractions under long-term continuous cropping. The size of the circle indicates the size of the Pearson correlation coefficient, and the larger the circle, the larger the coefficient. EC, electrical conductivity; AN, available nitrogen; and SOM, soil organic matter. * and ** indicate significant correlation at $p < 0.05$ and 0.01 levels, respectively.

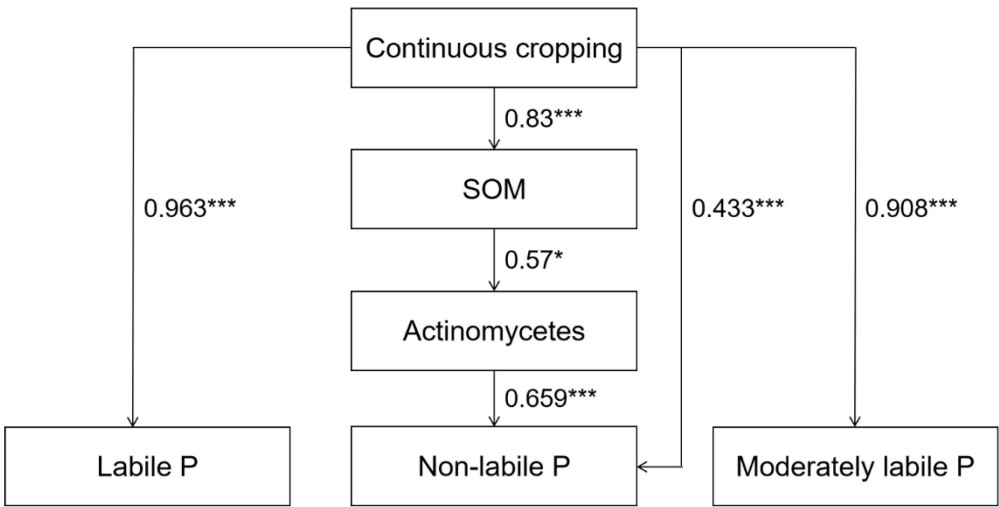

**Figure 8.** Structural equation model (SEM) of the relationship between the soil chemical properties, microbial community composition, and P fraction under long-term continuous cropping. The black arrow represents the path direction, the solid black line represents the positive significant correlation, and there is no negative correlation in this model. The numbers on the arrows indicate the strength of the standardized path coefficients, and the $R^2$ value represents the proportion of variance explained by each endogenous variable. * and *** indicate significant correlation at $p < 0.05$ and 0.001 levels, respectively.

## 4. Discussions

### 4.1. Phosphorus Accumulation in the Soil of Solar Greenhouses

Our study found that after 21 rounds of planting, the soil TP content in the soil was 3.41 times that in the 1st round, which is supported by the findings of Fu et al. [24], who also reported an increasing trend in soil TP with rounds of cultivation. The content of soil AP increased with an increase in planting rounds, but there was no significant difference between the 13th and 21st rounds (Figure 1). Some researchers believe that the soil AP leaching threshold is between 60–90 mg·kg$^{-1}$ [25–27]; the soil AP content was higher than the soil leaching threshold after the 9th round of planting in this experiment. These results indicate that long-term continuous cropping can increase the risk of P migration and leaching loss in the soil, and reduce the ability of soil TP to be converted to AP. This is inconsistent with the law of field soil [28]. This phenomenon may be caused by the excessive accumulation of total P in soil, owing to the habit of applying large amounts of mineral fertilizer and manure in solar greenhouses. Part of the AP migrates downward with the water, which may be related to the increasing water demand of cucumber plants due to the increasing temperature during spring cropping [29].

In this study, it was found that under continuous greenhouse cropping of cucumber, the absolute content of labile P and moderately labile P in the soil increased significantly with an increase in the number of planting rounds (Figure 2). This is consistent with previous findings that all P fractions in cabbage soil increased after 15 years of planting [30]. After chemical P fertilizers and manure are applied to the soil, a large amount of labile P is released, which preferentially accumulates in the soil. In comparison to chemical fertilizers, manure contributes more to the accumulation of labile $P_i$. Moreover, the planting environment of greenhouse vegetables is relatively closed, the amount of fertilizer input is large, and the utilization rate of P fertilizer is low. Thus, under the conditions of high temperature and high humidity, and long-term continuous fertilization and high-intensity continuous planting, the accumulation of labile P and moderately labile P in the soil becomes more serious [31,32].

Resin-$P_i$ is the most easily absorbed and utilized P component, with the strongest activity. NaHCO$_3$-$P_i$ is available P, which is beneficial for crop absorption and has strong activity. NaHCO$_3$-$P_o$ is a P fraction that can be absorbed by crops in the short term and

easily mineralized [33]. In this study, the relative distribution of P fractions showed that there was no significant difference in the proportion of soil resin-$P_i$ and $NaHCO_3$-$P_o$, while the proportion of $NaHCO_3$-$P_i$ decreased significantly after long-term continuous monocultivation. Although the absolute content of $NaHCO_3$-$P_i$ has been increasing, the increase in $NaHCO_3$-$P_i$ is less than that of other P fractions in the same period, so the proportion of $NaHCO_3$-$P_i$ also decreases. $NaOH$-$P_i$ and $NaOH$-$P_o$ are also considered to be P combined with iron and aluminum, which cannot be utilized and absorbed by crops in the short term, and the effectiveness of P is poor [34,35]. The results show that with an increase in continuous cropping rounds, the proportion of $NaOH$-$P_i$ in the cucumber greenhouse soil first decreases and then increases, and the proportion of $NaOH$-$P_o$ increases and then decreases (Table S4). The increase in $NaOH$-$P_i$ may be related to the long-term application of chicken manure, which has rich carbon sources in manure, leading to a large accumulation of SOM in soil at the later stage of continuous cropping, and the increase in P-saturated adsorption sites in soil particles [36,37]. Meanwhile, manure also releases organic acids to produce the chelation of Fe-P and Al-P. Because of this trade-off in the P component, as the $NaOH$-$P_i$ ratio increases, the $NaOH$-$P_o$ ratio decreases [38]. The dil. $HCl$-$P_i$ is mainly related to the primary apatite minerals and it is a calcium-bound compound. Dil. $HCl$-$P_i$ and $NaOH$-P, which are moderately labile P, can be used as the supplement of available P [39]. Continuous cropping significantly increased the ratio of Dil. $HCl$-$P_i$ to TP, which was expected because the soils studied had a neutral pH, high soil alkali saturation, and $Ca^{2+}$ as the main exchangeable cation [40]. In this experiment, under 21 rounds of continuous cropping, there was more soluble Ca in the soil solution to promote the formation of calcium and P. Conc. HCl-P ($HCl$-$P_i$ and $HCl$-$P_o$) is a slow-release phosphorus, which is difficult to be absorbed and utilized by plants, and can only be transformed and utilized by crops under special circumstances [41]. After long-term continuous cropping, the proportion of conc. $HCl$-$P_i$ in the soil gradually decreased, and the proportion of conc. $HCl$-$P_o$ gradually increased, indicating that, after long-term continuous cropping, the slow-release P in the soil gradually transformed into a more effective form. Residual-P is closed storage $P_i$, mainly a mixture of insoluble calcium P and humified $P_o$ [42]. The contents and proportions of these components decreased in the late continuous cropping period, suggesting that these components may be involved in the long-term P cycle. Soil residual-P tends to saturate after long-term fertilization, so these residual-P may change into labile or moderately labile P over time. In general, after long-term continuous cropping, the increase in labile P and moderate labile P may exacerbate the risk of P loss. Therefore, how to effectively utilize the labile P and moderate labile P accumulated in the soil, improve the utilization efficiency of P, and reduce the risk of P loss is the focus of future research.

### 4.2. Effects of Continuous Cropping on Soil Microbial Community Structure

PLFAs are widely used in microbial ecology to indicate the size and composition of microbial communities [43,44]. The PLFAs of greenhouse cucumber soil exhibited will experience obvious changes under long-term planting. Previous studies have shown that the most abundant types of microflora in healthy soils are bacteria, followed by fungi [45,46]. In this study, the contents of actinomycetes, $G^+$, $G^-$, and total PLFAs were significantly increased in the early stages of continuous cropping (rounds 1–9) and significantly decreased in the late stages of continuous cropping (rounds 13–21) (Table 3). After long-term continuous cropping, soil fungi and F/B increased significantly. The main reason for the above phenomenon may be that the soil nutrients of greenhouse cucumbers in the early planting period are relatively scarce, and soil nutrients become more abundant with the increase in planting time after planting. In the late planting period, soil nutrients are affected, thus affecting the change in microbial community composition (Table 3). In other words, short-term continuous monoculture is beneficial to increase the abundance of the soil microbial community, whereas a long-term continuous monoculture can change the dominant soil microorganisms, leading to a significant increase in the abundance of fungi.

$G^-$ bacteria are more active heterotrophs than $G^+$ bacteria [47]. However, with an increase in the planting time, the availability of soil nutrients did not increase, and $G^+$ bacteria became more competitive under the condition of limited nutrient availability [48]. Thus, a systematic decrease in the available nutrients may lead to an increase in $G^+/G^-$ ratios.

*4.3. Correlation Analysis of the Soil P Fractions, Soil Microbial Community Structure, and Soil Chemical Properties*

Changes in the soil nutrient content play an important role in the formation of the microbial community. In this study, SOM was found to be a key limiting factor for soil microorganisms (Figure 5). After long-term continuous cropping, a large amount of SOM was enriched, which is different to the soil chemical properties of field crops after continuous cropping. This may be related to the habit of applying large amounts of chemical fertilizers and manure in the solar greenhouse. Chicken manure has a high concentration of organic matter, but it is not fully mineralized and utilized during the current crop season, resulting in the accumulation of SOM with increasing planting time. For soil microbial communities, soil bacterial community diversity and composition were more sensitive to the combined application of manure and chemical fertilizers [49,50]. Actinomycetes were found to be the key factor affecting the content of moderately labile $P_o$ and non-labile P in soil at a depth of 0–20 cm soil (Figure S1). Long-term continuous cropping would reduce soil actinomycete abundance, leading to a decrease in the ratio of NaOH-$P_o$ to total P in the soil at the later stages [51].

**5. Conclusions**

In conclusion, the long-term continuous cropping of cucumbers in a solar greenhouse resulted in soil nutrient enrichment, and AP and TP reached the rich level. Long-term continuous cropping increased soil labile P, but had no significant effect on the ratio of labile P to TP, and there was no significant difference in P uptake in late continuous cropping. The results show that long-term continuous cropping can lead to a greater P accumulation in the soil. SEM analysis shows that, in addition to the direct effects of continuous cropping on soil P accumulation, SOM enrichment in the solar greenhouse also significantly affects the abundance of actinomycetes, which indirectly leads to the accumulation of non-labile P in the soil. This study clarifies the characteristics of soil P fraction accumulation after long-term continuous cropping, and provides a theoretical basis for future soil P fertilizer management and sustainable vegetable production. The dynamic balance between accumulated P and labile P should be taken into account in the future to avoid the excessive application of P fertilizer and manure, resulting in a waste of P resources.

**Supplementary Materials:** The following supporting information can be downloaded at: https://www.mdpi.com/article/10.3390/horticulturae8040320/s1, Table S1: Variations in soil basic properties with increasing continuous cropping rounds in the greenhouse soil. Different letters indicate significant difference among the treatments on $p < 0.05$; Table S2: Changes of soil inorganic phosphorus and organic phosphorus contents and relative distribution of P after long-term continuous cropping. Different letters indicate significant difference among the treatments on $p < 0.05$; Table S3: The relative distribution of P after long-term continuous cropping. Different letters indicate significant difference among the treatments on $p < 0.05$; Table S4: The contents of P fractions and relative distribution of P after long-term continuous cropping. Different letters indicate significant difference among the treatments on $p < 0.05$; Table S5: The relative distribution of $P_o$ after long-term continuous cropping. Different letters indicate significant difference among the treatments on $p < 0.05$. Figure S1: Pearson correlation analysis for PLFA and soil basic properties under long-term continuous cropping. EC, electrical conductivity; TP, total phosphorus; TK, total potassium; AN, available nitrogen; AP, available phosphorus; AK, available potassium; SOM, soil organic matter.

**Author Contributions:** Conceptualization, T.B., H.F. and Z.S.; methodology, T.B. and S.Z.; software, T.B. and S.W.; validation, T.B., H.F. and Z.S.; formal analysis, T.B. and X.L. (Xiao Li); investigation, T.B., S.Z., S.W., X.Z., Z.W., X.L. (Xiaoxia Li) and X.L. (Xiao Li); data curation, T.B.; writing—original

draft, T.B.; writing—review and editing, T.B., H.F. and Z.S.; visualization, T.B.; supervision, T.B. and Z.S.; project administration, T.B.; funding acquisition, Z.S. All authors have read and agreed to the published version of the manuscript.

**Funding:** This study was financially supported by the National Key R&D Program of China (2020YFD1000300) and China Agriculture Research System of MOF and MARA: CARS-23.

**Institutional Review Board Statement:** Not applicable.

**Informed Consent Statement:** Not applicable.

**Data Availability Statement:** The data presented in this study are available on request from the corresponding author.

**Acknowledgments:** The authors would like to thank Muhammad Awais for his contributions in revising the manuscript.

**Conflicts of Interest:** The authors declare no conflict of interest.

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
