# Peer review of "The Variation of Soil Phosphorus Fractions and Microbial Community Composition under Consecutive Cucumber Cropping in a Greenhouse"

_horticulturae, doi:10.3390/horticulturae8040320_

Round 1

Reviewer 1 Report

Dear authors,

The theme of the paper tittle “The variation of soil phosphorus fractions and microbial com-2 munity composition under consecutive cucumber cropping in 3 greenhouse” is very interesting and important in agronomy fields in a situation of P crisis and its lack as a resource. The content of P in soil is important to make a sustainable fertilization with lower rates. I really like the paper, the writing is very correct and easy for the reader but I have some comments and recommendations to improve the quality of this work.

Major concerns:

  1. Introduction section is longer in some parts, I think it would be easier for the reader to shorten it
  2. I don't understand why it only takes 4 rounds if it has much more data, it would be interesting to see the complete evolution or at least have some more data.
  3. I think it would be interesting to have data on the extraction of P by the crop to have a balance of P, is this possible?
  4. Chicken manure is being applied continuously, which after repeated applications is changing the microbiology of the soil, not only the P content influences that microbial composition but also the applied product.
  5. It would have been interesting to know the PLFAs of the manure that is used
  6. The soil P level is well above the critical level, so the soil microorganisms are not forced to mineralize organic P, I think that the effect of the changes in the activities and PLAFs are due to the applications continues manure, and for the continuous cultivation of cucumber, which is well reported by other works.Minor concerns:

Line 53: write full name the first time it is named in the text, review this throughout the manuscript

Line 80: Wu reported maximum increase, please indicated Wu et al (year)…

Fig 2: on the y-axis and cannot be mg P in PT

Section 5: change name “conclusion”

Line 483: “dissolve orthophosphates in inorganic and organic compounds”, in organic P is impossible, needs a better explanation

Table 2: manure no mature

Reviewer 2 Report

A very interesting topic for research work.
The abstract is written in clear and understandable language, with all the necessary elements.
material and methods described in an exhaustive way and allowing for an in-depth analysis of the experience.
test methods selected correctly, typical for this type of research. the analyzes were conducted very broadly.
one can write a subsection of statistical analyzes. The results are widely presented, so this section can be extended with more detailed descriptions of selected statistical methods / analyzes.
Results chapter very well presented, high level of research results presented. rich graphics and charts.
chapter 5 should be called instead of conclusion
I recommend for printing

Round 2

Reviewer 1 Report

Dear authors,

The proposed changes in the manuscript entitled "The variation of soil phosphorus fractions and microbial community composition under consecutive cucumber cropping in greenhouse" have greatly improved the work. Questions have been answered extensively and manuscript can be accepted for publication.